# Effect of Carbon Content and Intercritical Annealing on Microstructure and Mechanical Tensile Properties in FeCMnSiCr TRIP-Assisted Steels

Enzo Tesser [1,2], Carlos Silva [3,4], Alfredo Artigas [1] and Alberto Monsalve [1,*]

[1] Metallurgical Engineering Department, University of Santiago, Santiago 8320000, Chile; enzo.tesser@usach.cl (E.T.); alfredo.artigas@usach.cl (A.A.)
[2] Programs, Research and Development Directorate, Chilean Navy, Valparaíso 2360035, Chile
[3] Materials Chemistry Department, Faculty of Chemistry and Biology, University of Santiago, Santiago 9160000, Chile; carlos.silva@usach.cl
[4] Soft Matter Research and Technology Center, SMAT-C, Santiago 9160000, Chile
[*] Correspondence: alberto.monsalve@usach.cl; Tel.: +56-968475721

**Abstract:** Four TRIP (Transformation Induced Plasticity) assisted steels, three TBF (TRIP Bainitic Ferrite) steels and one TPF (TRIP Polygonal Ferrite) steel, were manufactured from three different carbon contents (0.2, 0.3 and 0.4 wt.% C), to study the evolution of their microstructure and tensile mechanical properties in 15 mm thick plates. TBF steels were subjected to the same austenitization heat treatment and subsequent bainitization isothermal treatment. The TPF steel was subjected to an intercritical annealing and subsequent isothermal bainitization treatment. All were microstructurally characterized by optical, scanning electron and atomic force microscopy, as well as X-ray diffraction. Mechanically, they were characterized by the ASTM E8 tensile test and fractographies. For the TBF steels, the results showed that when the carbon content increased, there were an increase in volume fraction of retained austenite, of the microconstituent "martensite/retained austenite" and in the tensile strength; and a decrease in the volume fraction of bainitic ferrite matrix and elongation; with an improvement in TRIP behavior due to the increase in retained austenite. The TPF steel presented around 50% ductile polygonal ferrite developing better TRIP behavior than the TBF steels. The evolution of the fractographies was ductile to brittle for TBF steels with an increase in carbon content, and for TPF, the appearance of the fracture surface was ductile.

**Keywords:** TRIP steels; retained austenite; martensite; bainite; ferrite; tensile strength

## 1. Introduction

Starting in 1980, due to the demands imposed by the automotive industry, a research trend developed that sought to improve the mechanical properties of steels, not only with the variation of chemical composition to produce interstitial or substitutional solid solution hardening, but also sought the variation of its microstructure through heat treatments that generated phase transformations, developing steels with a greater amount of alloying, polyphasic and very resistant elements called "Advanced High Strength Steels" or AHSS. These, apart from ferrite and pearlite, gave way to the presence of other phases and microconstituents such as bainite, martensite and retained austenite. In this context, and even though its formal discovery occurred around 1970 [1], steel with TRIP behavior (Transformation Induced Plasticity) gained importance in the industry since 1990, due to its condition of not excessively alloyed steel, high tensile strength, and good formability, attractive for the manufacture of automotive structures that need to absorb large amounts of energy when deforming [2–4]. The advantage of TRIP-assisted steels is that their elongation and mechanical strength increase as the retained austenite transforms into martensite during plastic deformation. This appropriate balance comes from the strain-induced transformation of retained austenite to martensite, during plastic strain. The

stabilization of austenite at room temperature is due to the carbon enrichment that occurs during the specific thermomechanical treatments carried out during the manufacture of these steels [5,6]. Other parameters that influence the stability of austenite are its grain size, the stress state of the surrounding matrix, and temperature [7].

In understanding the mechanical behavior of this type of steels, a precise characterization of the microstructure, which is normally multiphasic [8,9], is of great importance. Based on this idea, steels were first developed with a chemical composition (0.15–0.4) wt.% C—(1.5–2.5) wt.% Si—(1.5–2.5) wt.% Mn, which have a polygonal ferrite matrix, achieved by intercritical annealing, and an abundant amount of carbide-free bainitic ferrite plus retained austenite and a minor amount of residual martensite, achieved with bainitic isothermal treatment. Thus, for example, it is known that the plastic forming properties are determined by the characteristics of the retained austenite [10,11]. The named TRIP polygonal ferrite or TPF steel has attained extremely large total elongation up to 30–40% due to the strain induced transformation in a large strain range. The tensile strength of this steel could not exceed 980 MPa, because the matrix structure is based on soft polygonal ferrite due to intercritical annealing. However, when matrix structure is replaced, for example, with bainitic ferrite, martensite, or these combined structures, the yield stress and tensile strength of the steel could enhance up to 980–1470 MPa by maintaining a good stretch flangeability. Such is the case of steels named TRIP bainitic ferrite or TBF and TRIP martensitic or TM [12]. If the TBF steels are applied to relatively large forging parts, high hardenability may be required to obtain the mixed microstructure of bainitic ferrite and metastable retained austenite. In general, hardenability of the steel is improved by the addition of alloying elements such as Cr, Mo, Ni, Mn, B, etc. However, there is no research investigating the effects of hardenability on microstructure and mechanical properties in the hot-forged medium-carbon TBF steels [13].

TRIP effect increases the homogeneous strain, so it is expected to have acceptable formability. Nevertheless, most important in formability of metals and alloys are the normal anisotropy index r (always known as Lankford coefficient) and the planar anisotropy index ($\Delta r$). However, the focus of this work is heavy industry, where formability is not the mean desired mechanical property.

Therefore, the novelty and motivation of the present work consists of exploring the applicability of these steels in structural applications such as mining, shipbuilding, maritime and port infrastructure, where large thicknesses are usually used, and formability is not as relevant as toughness.

In this study, the effect of carbon content and intercritical annealing on microstructure and tensile mechanical properties in FeCMnSiCr TRIP-assisted steel (TPF and TBF) 15 mm thickness sheet will be evaluated.

## 2. Materials and Methods

### 2.1. Base Material Manufacturing

Three different steels (S1, S2, S3) were prepared in form of 30 Kg ingots by induction melting process, with C, Mn, Si and Cr as principal alloy elements. The criterion for choosing steel chemical composition was, based on typical TRIP steels, to increase the amount of carbon in order to evaluate its effect on the morphology of the different phases and micro-constituents achieved by different heat treatments. In addition, with other alloying elements, mainly Chromium and Nickel, to achieve a better hardenability of the steels, since one of the objectives of this research is to work with 15 mm thick plates, oriented to use in industries other than the automotive one. The ingots were hot forged into slabs and then heated to 1473 K (1200 °C) and hot rolled into plates (15 mm in thickness) with the finishing rolling temperature being 1123 K (850 °C). The plates were cooled in air to room temperature. The final chemical composition of steel plates was determined by optical emission spectroscopy (OES) and are shown in Table 1.

**Table 1.** Chemical Compositions by OES (wt.%).

| Steels | C | Mn | Si | Cr | Al | Cu | Mo | Ni | P | S | Fe |
|---|---|---|---|---|---|---|---|---|---|---|---|
| S1 | 0.166 | 1.872 | 1.534 | 0.242 | 0.343 | 0.0995 | 0.017 | 0.050 | 0.005 | 0.022 | Bal. |
| S2 | 0.285 | 1.829 | 1.445 | 0.242 | 0.068 | 0.0873 | 0.017 | 0.049 | 0.005 | 0.021 | Bal. |
| S3 | 0.397 | 1.920 | 1.470 | 0.462 | 0.008 | 0.0873 | 0.013 | 0.481 | 0.016 | 0.019 | Bal. |

*2.2. Thermal Study and Heat Treatments*

Once the base material (S1, S2, S3) was obtained with a suitable chemical composition for steels with TRIP behavior, a thermal study was carried out to obtain the main working temperatures ($A_1$, $A_3$, Bs, Ms) and subsequently design the optimum heat treatment to generate TPF and TBF steels. The $A_1$ and $A_3$ temperature were measured using bibliographic equations data (BBL), differential scanning calorimetry (DSC), differential thermal analysis (DTA) and study of the microstructure evolution with optical microscopy (OM) and software imaging (IMG) which consisted of annealing at different temperatures and times in the biphasic field ($\alpha + \gamma$) on cubic samples of 15 mm edge and subsequently quenched them in water. The samples were cut into two parts with abrasive disc, polished and etched with a 3% Nital solution and analyzed by OM [14–16], quantifying the amount of polygonal ferrite and martensite from the transformed austenite determining the percentages of different colored areas by Image-Pro Plus software Figure 1.

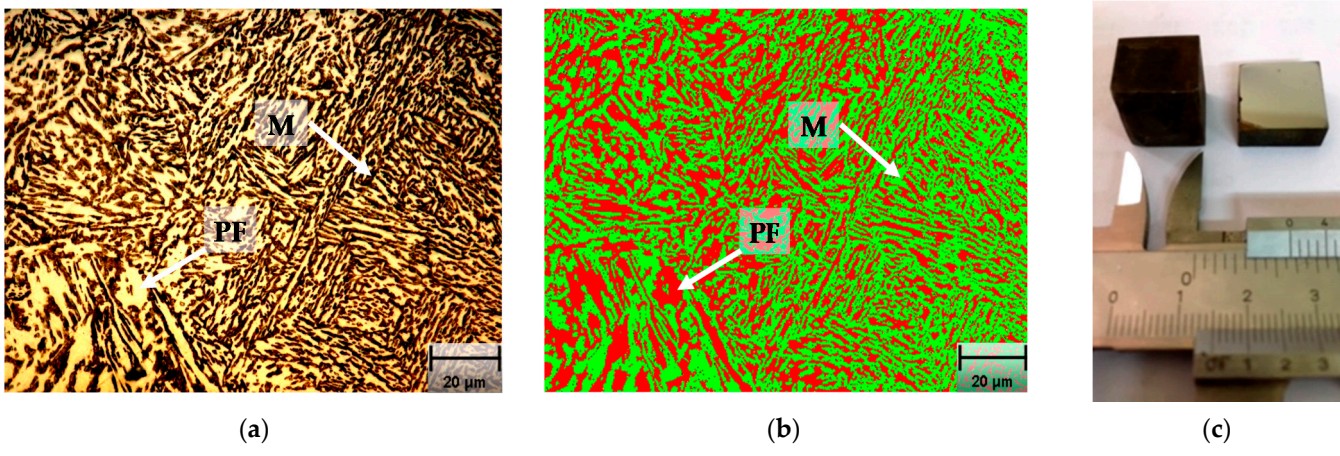

| (**a**) | (**b**) | (**c**) |
|---|---|---|

**Figure 1.** Micrography example showing the quantification procedure by Image-Pro Plus software for the polygonal ferrite (PF) and martensite (M) in S1 sample, annealed to 760 °C and then quenched in water: (**a**) micrography polished and etched with a 3% Nital solution where the bright color is PF and dark color is M; (**b**) the same micrography quantified, where PF is red (50 vol.%) and M is green (50 vol.%); (**c**) cubic sample of 15 mm edge and next cut in half and polished.

The Bs and Ms temperatures were found using only BBL [17]. For the case of TPF steel, the optimal intercritical annealing temperature to achieve around 50% PF was determined by the same procedure as the microstructure evolution with optical micrography, just in S1 steel, because its carbon wt.% is optimal to achieve a good quantity of PF (760 °C). In the case of TBF steels, there is no intercritical annealing, therefore a temperature above $A_3$ was chosen by 50 °C or more (910 °C), to ensure the maximum dissolution of carbides at a not excessively high temperature. The optimal holding time determined for intercritical annealing and austenitizing annealing was 1200 s. The Table 2 shows the measured temperatures by thermal study.

**Table 2.** Measured temperatures by thermal study in Celsius degree. BBL: bibliographic data. DSC: differential scanning calorimetry. DTA: differential thermal analysis. IMG: microstructure evolution analysis with OM. $A_1$: critical temperature 1. $A_3$: critical temperature 3. Bs: bainite start temperature. Ms: martensite start temperature. 50/50: $50\%\alpha + 50\%\gamma$ temperature.

| Steel | Source | $A_1$ | $A_3$ | Bs | Ms | 50/50 |
|-------|--------|-------|-------|-----|-----|-------|
|       | BBL    | 728   | 866   | 581 | 389 | -     |
| S1    | DSC    | 713   | 855   | -   | -   | -     |
|       | DTA    | 725   | 855   | -   | -   | -     |
|       | IMG    | 712   | 865   | -   | -   | 760   |
|       | BBL    | 725   | 835   | 556 | 343 | -     |
| S2    | DSC    | 728   | 820   | -   | -   | -     |
|       | DTA    | 717   | 825   | -   | -   | -     |
|       | IMG    | 715   | 830   | -   | -   | -     |
|       | BBL    | 720   | 796   | 473 | 228 | -     |
| S3    | DSC    | 730   | 822   | -   | -   | -     |
|       | DTA    | 727   | 812   | -   | -   | -     |
|       | IMG    | 720   | 815   | -   | -   | -     |

After achieving the temperatures and annealing times for the possible TPF and TBF steels, it is necessary to maximize the volume quantity of retained austenite (RA), optimize its carbon enrichment, and minimize the volume quantity of precipitated carbides. For this, the temperatures obtained in Table 2 were used to generate a series of 9 different heat treatments (Table 3), obtaining the optimal bainitic isothermal treatment (austempering) temperature and time, analyzing the different samples by X-ray diffraction (XRD) and OM. For the different isothermal treatments, a salt furnace was used.

**Table 3.** From left to right heat treatments for TPF and TBF steels. IA: Intercritical Annealing; IT: Isothermal Treatment (Austempering); AA: Austenitizing Annealing; T: temperature; t: time.

| Heat Treatments for TPF Steels (S1) | | | | Heat Treatments for TBF Steels (S1, S2, S3) | | | |
|---|---|---|---|---|---|---|---|
| N | T (°C) IA | t (s) IA | T (°C) IT | t (s) IT | N | T (°C) AA | t (s) IA | T (°C) IT | t (s) IT |

| N | T (°C) IA | t (s) IA | T (°C) IT | t (s) IT | N | T (°C) AA | t (s) IA | T (°C) IT | t (s) IT |
|---|-----|------|-----|------|---|-----|------|-----|------|
| 1 | 760 | 1200 | 350 | 600  | 1 | 910 | 1200 | 350 | 600  |
| 2 | 760 | 1200 | 350 | 1000 | 2 | 910 | 1200 | 350 | 1000 |
| 3 | 760 | 1200 | 350 | 1500 | 3 | 910 | 1200 | 350 | 1500 |
| 4 | 760 | 1200 | 400 | 600  | 4 | 910 | 1200 | 400 | 600  |
| 5 | 760 | 1200 | 400 | 1000 | 5 | 910 | 1200 | 400 | 1000 |
| 6 | 760 | 1200 | 400 | 1500 | 6 | 910 | 1200 | 400 | 1500 |
| 7 | 760 | 1200 | 450 | 600  | 7 | 910 | 1200 | 450 | 600  |
| 8 | 760 | 1200 | 450 | 1000 | 8 | 910 | 1200 | 450 | 1000 |
| 9 | 760 | 1200 | 450 | 1500 | 9 | 910 | 1200 | 450 | 1500 |

The volume fractions (vol.%) of the RA phase was quantified from the integrated intensity of the $(110)\alpha$, $(200)\alpha$, $(111)\gamma$, $(200)\gamma$, and $(220)\gamma$ peaks, obtained via XRD using Cr-K$\alpha$ radiation [18]. The equipment used was a Rigaku MiniFlex diffractometer (Rigaku, Tokyo, Japan), with continuous scanning mode at a speed of 5 deg/min in a range of 30 to 140 deg. To record the scattered x-rays, a high-speed Rigaku D/TeX Ultra detector (Rigaku, Tokyo, Japan) was used. To minimize instrumental and experimental errors, the diffractometer was calibrated according to the supplier's recommendations, being the error around 3–4%, in accordance with ASTM E975. Then drift and broadening of the instrument's own peaks were calculated using a LaB$_6$ standard (NIST 660C). The measurements of the samples were corrected assuming Lorentzian (Cauchy) profile. Using the same profiles, the phases present in the samples were identified. The carbon concentrations in RA of the specimens

were estimated from Onink Equation (1) [19]. The lattice constant was determined from the (111)γ, (200)γ, and (220)γ peaks of the Cr-Kα radiation.

$$C\gamma = (a\gamma - 0.35550)/3.8 \times 10^{-3} \tag{1}$$

where Cγ is the carbon content (wt.%) in RA and aγ is the lattice constant of RA.

### 2.3. Microstructural Study

The microstructure of the different steels subjected to the chosen heat treatment was analyzed using optical microscopy (OM) with 1000× magnification, scanning electron microscopy (SEM) with 1000× and 5000× magnification, and atomic force microscopy (AFM) with Gwyddion analysis software (version 2.59, Czech Metrology Institute, Brno, Czech Republic), to determine the presence, morphology, and topography of polygonal ferrite (PF), bainitic ferrite (BF), martensite (M), and possible retained austenite (RA). All samples were etched with 3% Nital, to use its properties of optical reveal and differential chemical dissolution of phases and microconstituents. The nomenclature proposed by Zajac et al. [20], was used to classify the different morphologies of the phases and microconstituents present in the steels, related to complex bainitic microstructures.

### 2.4. Tensile Mechanical Properties

Tensile specimens were manufactured and tested following the procedures described in the ASTM E8M standard [21], to analyze the tensile mechanical properties and substantiate the TRIP effect of different steels. The chosen specimens for the test had their gauge length five times the diameter and were strained in a traction machine ZWICK ROELL Z050 (ZwickRoell Group, Ulm, Germany) with a capacity of 5 tons. The specimen diameter was 6 mm. Finally, the fracture surface of the samples was analyzed by SEM to determine the kind of tensile fracture and material behavior.

## 3. Results

### 3.1. Base Material

Figure 2 shows the results of the XRD for each of the samples subjected to the different heat treatments established in Table 3. When analyzing the vol.% RA (Figure 2a) it is appreciated that the heat treatment number 6 generates the highest percentages for each one of the steels, except for S1 (TBF), which is also over 10%. On the other hand, when analyzing the wt.% C in the RA (Figure 2b), the heat treatment 6 presents the most homogeneous amounts, ranging from 1.4 to 1.6%.

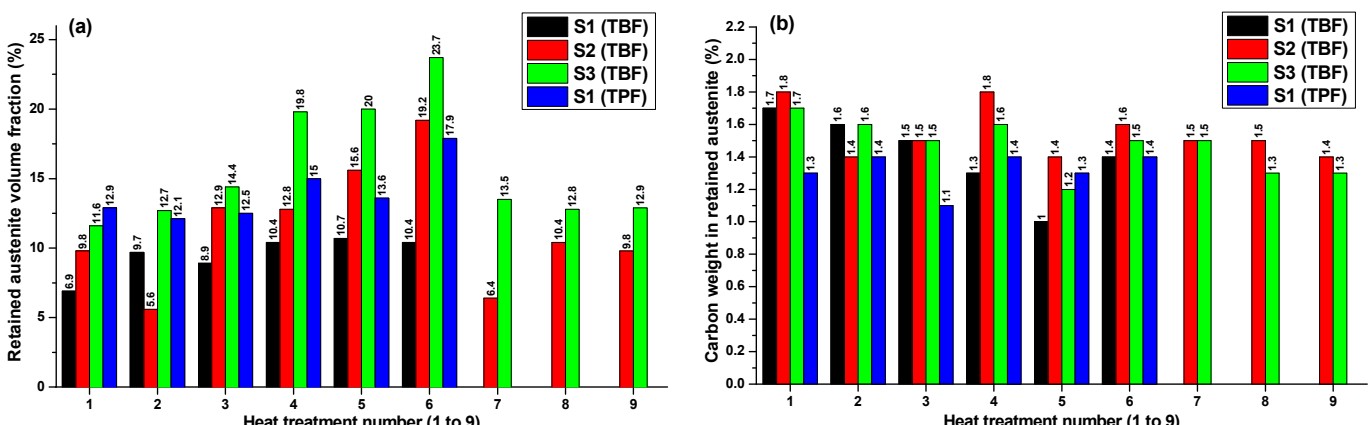

**Figure 2.** XRD results about all the heat treatments to find the optimal vol.% RA and its carbon enrichment (the exact amount appears at the top of each bar in the charts). (**a**) Retained austenite volume fraction and (**b**) Carbon weight percentage in retained austenite.

Therefore, it is established that the optimal treatment for S1, S2 and S3 to generate possible TPF and TBF steels, maximizing the vol.% RA, and optimizing its carbon enrichment, is number 6, generating route 1 for TBF and route 2 for steel TPF, as shown in Figure 3.

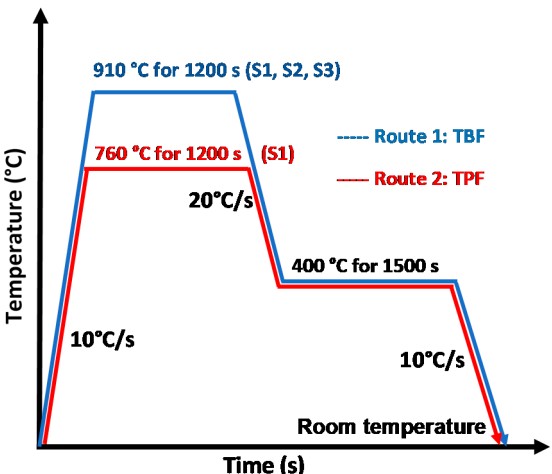

**Figure 3.** Heat treatments selected to achieve TPF and TBF steels and their respective routes.

The new designation for the possible TBF steels will be A, B, C, coming from the chemical compositions S1, S2, S3, respectively, with heat treatment route 1. While the new designation for the possible TPF steel will be D, coming from the chemical composition S1, with heat treatment route 2. The diffractograms in Figure 4, graphically show the difference between the steels achieved with heat treatment number 6, regarding to peaks relative intensities of austenite ($\gamma$). Table 4 shows a resume with the designations, heat treatment selected and its XRD parameters. Consistent with the increase in wt.% C, steels A, B and C increased the vol.% RA from 10.4 to 23.7%, while steel D achieved 17.9%. The stable wt.% C in the RA, between 1.4–1.6%, will be important to evaluate the capability of transformation by plastic deformation from RA to M and the tensile mechanical properties of the different steels (A, B, C, D) in the same condition.

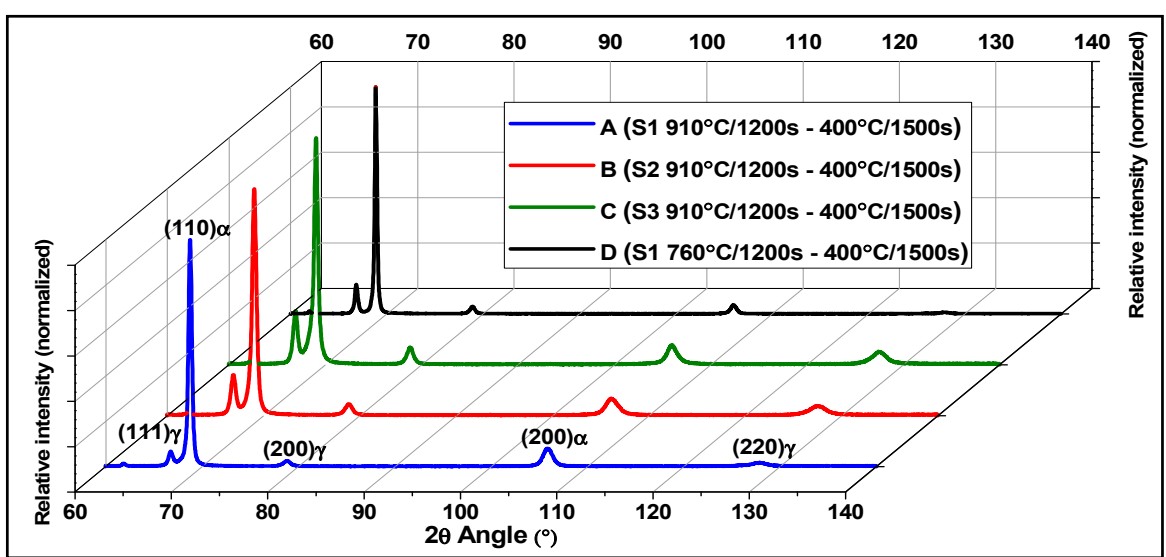

**Figure 4.** A, B, C, D, steels diffractograms, showing the difference between the steels, regarding to peaks relative intensities of $\gamma$.

**Table 4.** A, B, C, D, steels. A: annealing; IT: intercritical treatment; RA: retained austenite; α: BCC or BCT microconstituent; $a_0$: lattice parameter.

| Steels | | | Final Heat Treatment | | | | XRD Results | | | |
|---|---|---|---|---|---|---|---|---|---|---|
| | | | T (°C) A | t (s) A | T (°C) IT | t (s) IT | vol.% RA | vol.% α | $a_0$ (Å) RA | wt.% C RA |
| A | TBF | S1 | 910 | 1200 | 400 | 1500 | 10.4 | 88.3 | 3.609400 | 1.4 |
| B | TBF | S2 | 910 | 1200 | 400 | 1500 | 19.2 | 79.2 | 3.613900 | 1.6 |
| C | TBF | S3 | 910 | 1200 | 400 | 1500 | 23.7 | 74.2 | 3.613690 | 1.5 |
| D | TPF | S1 | 760 | 1200 | 400 | 1500 | 17.9 | 80.3 | 3.606854 | 1.4 |

## 3.2. Microstructural Evolution

### 3.2.1. Optical Microscopy

Figure 5 shows the results obtained for steels A, B, C, D by OM under light microscope at 1000× magnification, etched with 3% Nital. As a general concept, the Nital etch ferrite at a rate that varies with the crystal orientation of each grain relative to the plane of polish, which produces steps at grain boundaries and reflectivity differences, between grain or phase, which produces grooves, and therefore, reveals the ferrite grain boundaries. In light microscopy, ferrite appear like bright or white phase, like austenite and cementite. The exception is M, which appear darker than ferrite (grey or black phase), due to different corrosive attacks and roughness [22–24].

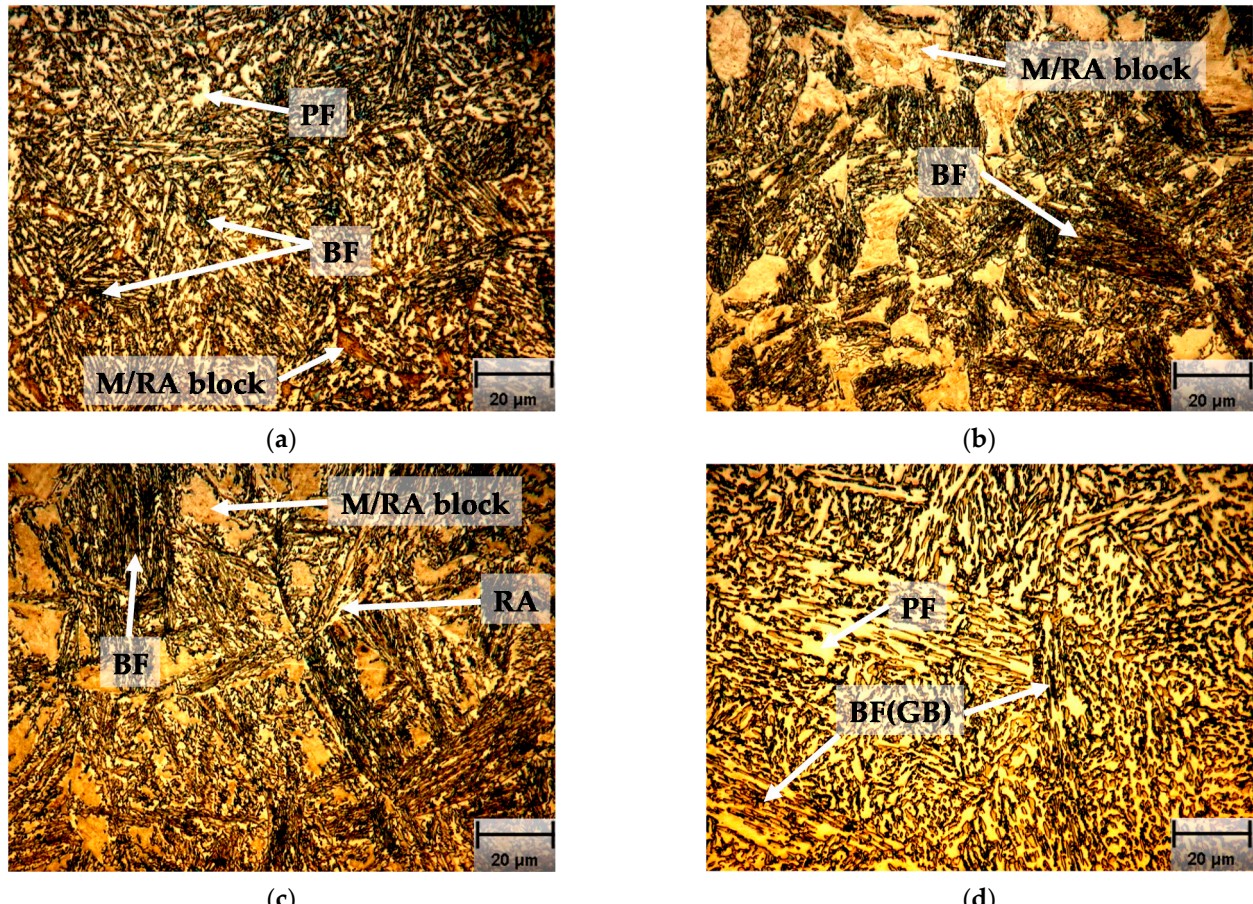

**Figure 5.** Optical micrograph of heat-treated steels. Results obtained for steels A, B, C, D by OM under light microscope at 1000× magnification, etched with 3% Nital. (**a**) A steel; (**b**) B steel; (**c**) C steel; (**d**) D steel.

In steel A (Figure 5a), the existence of a finely distributed white phase islands can be seen in a network of not well-defined and irregular black lines, in addition, the existence of randomly distributed brown islands is appreciated. Due to the amount of carbon in the steel (0.166 wt.%) and the heat treatment applied, the white phase should correspond to PF, the black lines to grain boundaries between BF, and the brown islands to M in block. Due to the morphology presented as a whole and knowing positively the existence of vol.% RA (10.4%), it can be supposed that this should be found between the BF plates and on the periphery of the block M islands, forming a characteristic microstructure of steels with TRIP behavior, which corresponds to the set formed by carbide-free BF, M and RA. Both the M and the RA will be presented together as a microconstituent (M/RA) difficult to differentiate between them, whose morphology (block or films) and their proportion will depend on the amount of carbon and the heat treatment applied, determined by temperatures, type of cooling (continuous or isothermal) and cooling rates [25]. Therefore, in A steel, just OM, it can establish the presence of PF, BF, and M/RA block.

Steel B (Figure 5b) shows a different appearance compared to steel A. It presents BF colonies and a greater quantity of M/RA blocks, the existence of PF is not observed. This steel has a higher amount of C (0.285 wt.%), determining a decrease in Bs and Ms, and generating a gradual lower transformation rate for the same austempered temperature (400 °C), when the transformation of the austenite to BF take place and it is gradually enriching with carbon. This could indicate that after the 1500 s of isothermal transformation, there would still be a considerable amount of carbon enriched austenite untransformed, which in the final cooling to room temperature gave rise to the islands of M/RA.

Steel C (Figure 5c) does not present a great difference compared to steel B. Colonies of BF and an appreciable amount of M/RA block are distinguished. Perhaps, the only difference could be a series of finely distributed phases lighter or brighter than the M/RA, which due to the fact of having been etched with Nital, could indicate the presence of PF or RA in blocks (phases revealed as "white").

Considering that steel C has higher amount of carbon (0.397 wt.%) and almost twice as much chromium than A and B (0.462 wt.%), added to 23.7 vol.% of RA, could indicate the possibility that these "white phases" were indeed RA in blocks or at least M/RA high in RA, which could be confirmed later by SEM.

Steel D (Figure 5d) is the one that presents the greatest morphological difference, since a coarser structure with the presence of PF and BF is observed, like "granules", in addition to a series little island that could be attributed to the M/RA microconstituent. This morphology could be identified as granular bainite (GB), however, its formation mechanism is controversial, because, as established by Bhadeshia, GB is a characteristic microconstituent of TRIP aided steels achieved just by continuous cooling treatment [26], and the heat treatment for all samples was produced with isothermal transformation. On the other hand, Slama et al. [27], reported GB from the "granularization" process of lath-like structures by isothermal treatment. This granularization process on the ferrite matrix consists in the disappearance of all acicular block boundaries (highly disoriented) and lath boundaries (lowly disoriented). This process is triggered by the presence of upper bainite and affects the entire microstructure, even those made of lower bainite or martensite. In this case, steel D was annealed in 760 °C by 1200 s, producing PF (50 vol.%) in biphasic field, enriching the remainder austenite with approximately 0.5–0.6 wt.% C. Then, with isothermal treatment (400 °C) the remainder austenite should been transformed first in lath-like bainite, but after 1500 s, should been aged and taken a granular shape.

### 3.2.2. Scanning Electron Microscopy (SEM)

Figure 6 shows the results obtained for steels A, B, C, D by SEM at 1000× and 5000× magnification etched with 3% Nital. In this case, the concept used for the identification of phases and/or microconstituents is the preference or the greater chemical dissolution that Nital causes over the ferrite compared to the rest. Therefore, the ferrite will be a dark phase and with greater depth than austenite, martensite, and carbides,

which will appear as brighter or lighter phases, and their differentiation can be made by their morphologies.

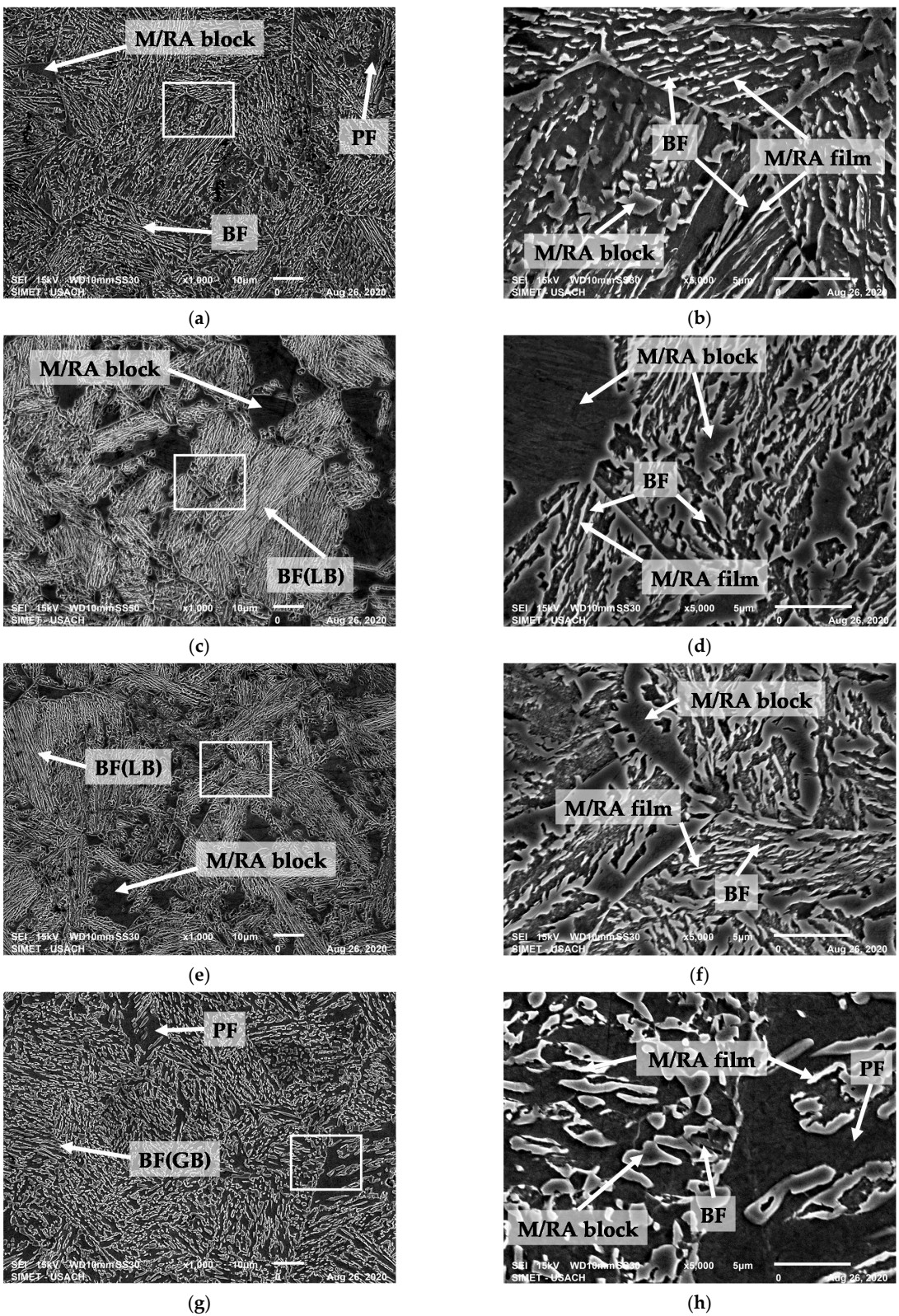

**Figure 6.** Scanning electron microscopy. Results obtained for steels A, B, C, D by SEM at 1000× (**a**,**c**,**e**,**g**) and 5000× (**b**,**d**,**f**,**h**) magnification etched with 3% Nital.

In steel A (Figure 6a), it is possible to distinguish a matrix of BF mostly in plates or laths with islands of M/RA in blocks and PF distributed in a random way. In addition, the prior austenitic grain boundaries can be seen, from which the BF plates nucleated and grew. In Figure 6b a zone of confluence of prior austenitic grain boundaries is more clearly shown, with its well-defined oriented BF plates. Among the BF plates, M/RA can be seen mostly in film, and minority in blocks, with brighter and whiter borders, attributable to possible RA presence.

Steel B and C in Figure 6c,e, respectively, show M/RA in blocks and some colonies of well-defined lath-like bainite (LB) by the M/RA film, and in Figure 6d,f, a coarser microstructure is appreciated with well-defined sheaves shape [28], with M/RA film and M/RA block between them. PF is not appreciated in these steels, confirming what was proposed in the OM analysis, regarding with that the "white phases" could correspond to RA in blocks or at least M/RA high in RA.

Steel D in Figure 6g,h shows PF, BF (with GB morphology) and, M/RA mostly as blocks and minority like films, with brighter and whiter borders attributable to possible RA presence, surrounded by irregular BF blocks. Due granularization process and, morphology change, from LB to irregular blocks, is difficult to distinguish between PF (proeutectoid) and BF (GB).

### 3.2.3. Atomic Force Microscopy (AFM)

Figure 7 shows the results obtained for steels A, B, C, D by AFM, etched with 3% Nital, where a,c,e,g, correspond to topographic images in 2 dimensions, while b,d,f,h are the same images seen in 3 dimensions. The concept used for the identification of phases and/or microconstituents is like used with SEM, exploiting the particularity of Nital to preferentially dissolve different phases and microconstituents present in multiphase steels, being able to generate topographic images that allow their morphology to be analyzed in detail.

The great advantage of the AFM compared with SEM is the possibility of imaging surfaces in different environments, without any vacuum or special sample treatment, with very high resolution [29]. The multiphase samples should present at least three phases: PF, RA, and BF (and possibly also M), but they are not easy to identify in a large AFM scan area. The BF can be identified according to the kinetics of formation during the isothermal treatment. The bainite transformation proceeds by repetitive events, leading to the formation of many sheaves of BF across the prior austenite grains, and a sheaf morphology is commonly observed. These BF laths (LB) are more resistant to attack by Nital than PF, as has been reported by Ros–Yañez et al. [30]. Through topographical analysis, three different levels of height can be identified in this type of sample. The lowest level, clearly corresponding to PF; an intermediate level, corresponding to BF; and the highest level, composed of M/RA. The surface roughness parameter was not used because the samples did not present conclusive results that could relate to the different phases and microconstituents present in the steels.

Analyzing only the relative heights, the steels A, B, C (Figure 7a–f) shows a topographic image with similar characteristics, where BF and M/RA (in block and films) can be identified, not appreciating the presence of PF. The morphology of BF is clearly like fine sheaves, characteristic of bainitic steels, however, as the amount of carbon from A to C steels increases, M/RA in block is coarser.

The case of steel D in Figure 7g,h is different because it shows PF, BF with GB morphology and M/RA, mostly as blocks and minority like films, surrounded by irregular BF blocks, such as was showed in SEM pictures. Due granularization process and, morphology change, from LB to irregular blocks, it is difficult to distinguish between PF and GB if it does not have a relative height parameter, but with AFM test, clearly it is appreciated a deeper area attributable to PF.

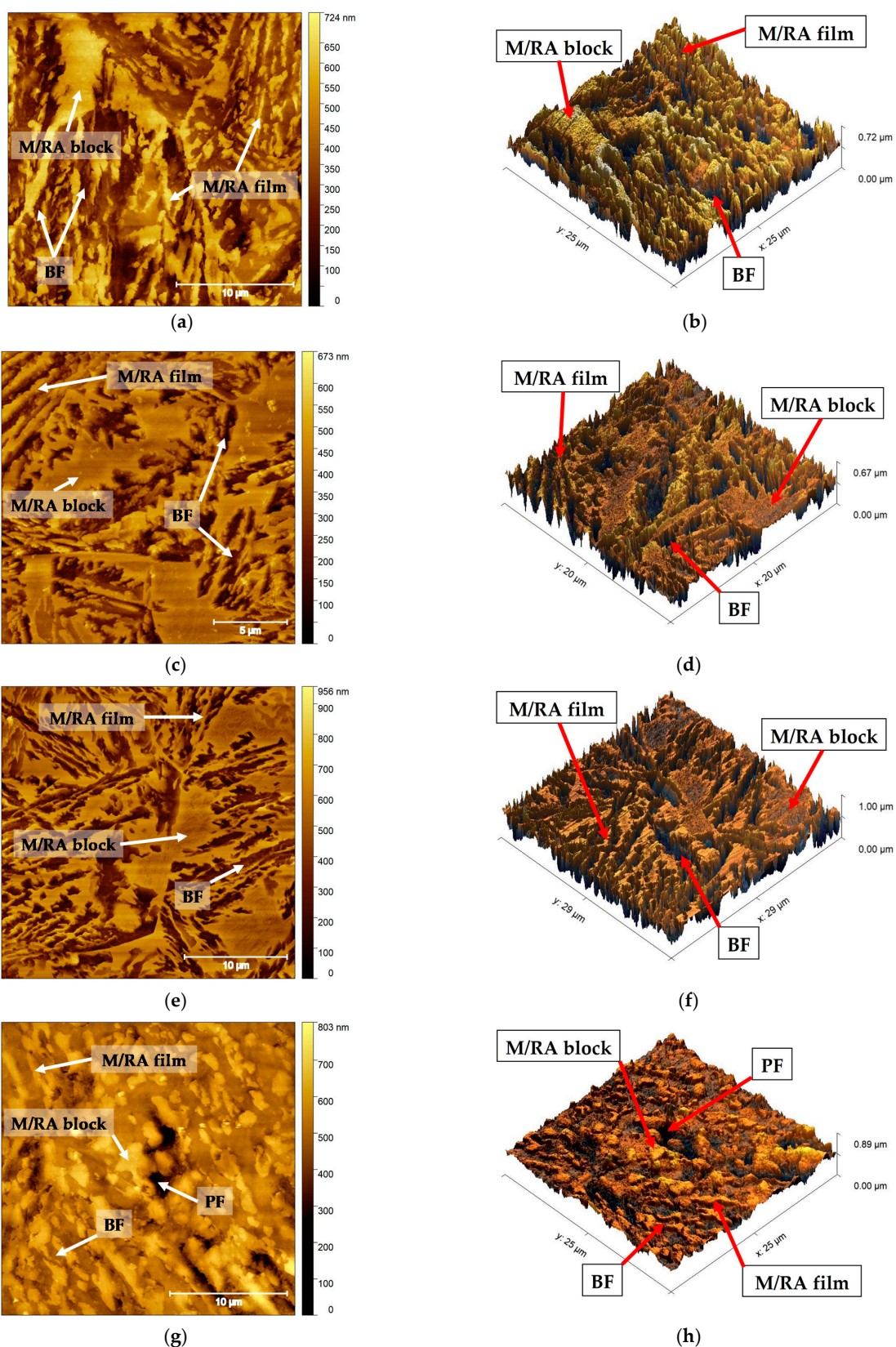

**Figure 7.** Atomic Force Microscopy. Results obtained for steels A, B, C, D by AFM, etched with 3% Nital, where (**a**,**c**,**e**,**g**) are topographic images in 2D, and (**b**,**d**,**f**,**h**) are the same images in 3D.

In Figure 8a,b, a topographic image in 2 and 3 dimensions of a smaller area of steel B is shown as an example. Thanks to the higher resolution of the AFM technique, the detail

of the bainitic morphology can be appreciated, thus representing a great alternative to explore bainitic steels, better than SEM and TEM. Figure 8c represents a profile of heights taken along Figure 8a, where the valleys are BF, with relative height close to 150 nm, and the peaks M/RA, with relative height close to 400 nm. The width of a BF (LB) is not more than 1 μm, just like M/RA films.

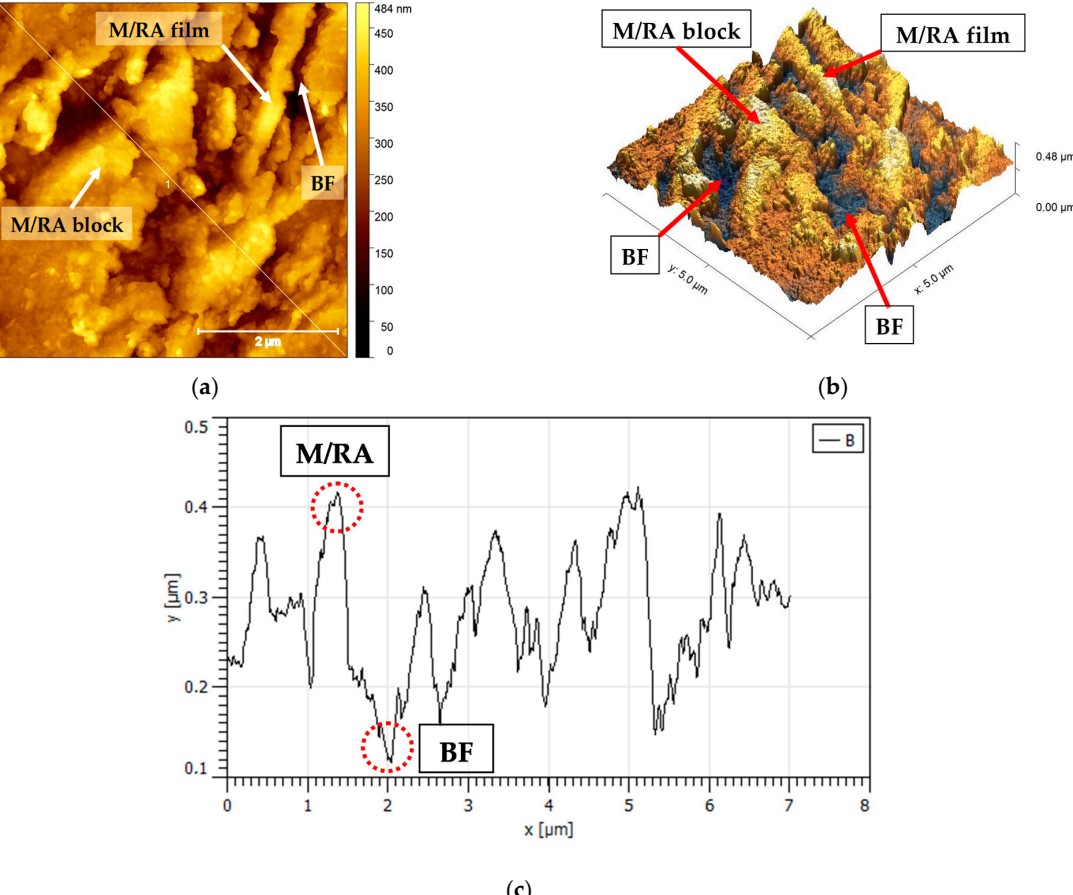

(a)

(b)

(c)

**Figure 8.** Atomic Force Microscopy. Results obtained for steel B. Topographic image, (**a**) 2D and (**b**) 3D, of a smaller area (5 μm²) showing the detail of the bainitic morphology; (**c**) represents a profile of heights taken along (**a**).

### 3.3. Tensile Test

#### 3.3.1. Mechanical Properties

Table 5 shows the results of the monoaxial, engineering and true tensile tests, and in Figure 9 their respective graphs of stress versus strain. The relationships used to compute true stress and true strain were:

$$\sigma_T = \sigma_e(1 + \varepsilon_e) \qquad (2)$$

$$\varepsilon_T = \ln(1 + \varepsilon_e) \qquad (3)$$

where $\varepsilon_e$ is the engineering strain; $\varepsilon_T$ is the true strain; $\sigma_e$ is the engineering stress and $\sigma_T$ is the true stress.

**Table 5.** Tensile test results. YS: Yield Strength; UTS: Ultimate Tensile Strength; FS: Fracture Strength; UE: Uniform Elongation; TE: Total Elongation; Exz: Area reduction; n: strain hardening index (Hollomon's equation); K: strength coefficient.

| STEEL | Engineering Tensile Test | | | | | | | True Tensile Test | | | | | |
|---|---|---|---|---|---|---|---|---|---|---|---|---|---|
| | YS MPa | UTS MPa | YS/UTS - | FS MPa | UE % | TE % | Exz % | YS MPa | UTS MPa | YS/UTS - | UE % | n - | K MPa |
| A | 735 | 913 | 0.81 | 713 | 10.7 | 18.4 | 35.6 | 735 | 1016 | 0.72 | 11.1 | 0.12 | 757 |
| B | 1020 | 1442 | 0.71 | 1375 | 11.5 | 14.3 | 12.9 | 1020 | 1615 | 0.63 | 11.6 | 0.17 | 1045 |
| C | 1060 | 1527 | 0.69 | 1511 | 12 | 12.1 | 0.9 | 1060 | 1710 | 0.62 | 11.4 | 0.18 | 1077 |
| D | 560 | 897 | 0.62 | 777 | 15.8 | 22.3 | 29.2 | 560 | 1047 | 0.53 | 16 | 0.23 | 559 |

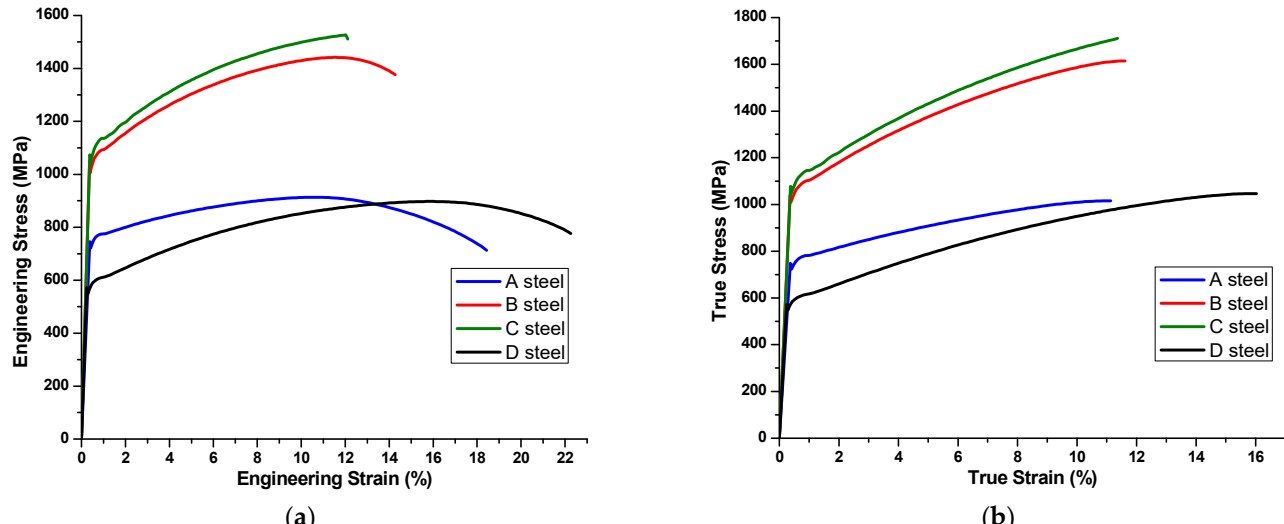

**Figure 9.** Tensile test graphs strain vs. stress. (**a**) Engineering graph. (**b**) True graph.

When comparing the yield strengths (YS) of steels A, B and C, which have the same heat treatment, clearly it can be seen that with the increase in the wt.% C, the value of YS increases. The steels with the lowest YS value are A and D, which have the same chemical composition, both with a low wt.% C, but between them there is a notable difference of 175 MPa, due to dissimilar heat treatments that produced different microstructures. Steel A presents a matrix of BF, high in density of dislocations, making it difficult to slide when elongated. Meanwhile, D steel has around 50% PF, with an evident lower density of dislocations, making it a material with a lower YS due to the greater ease of dislocations to slide.

Regarding Ultimate Tensile Strength (UTS), both engineering and true, the trend of increasing resistance is maintained with the increase in the wt.% C, highlighting the B and C steels with a higher comparative value. Remarkable is the behavior of steel D, since the resistance gap with respect to steel A, previously 175 MPa in YS, drops to 16 MPa for engineering UTS and 31 MPa for true UTS.

This is also evidenced in the YS/UTS relationship, which shows a factor that approaches to 1 when the difference between YS and UTS is minimal, and 0 when the difference is maximum. In this relationship, for steels A, B and C, which have the same heat treatment, the factor decreases with the increase in the wt.% C, while steel D is the one that presents the lowest YS/UTS factor compared to all steels, despite being the one with the least wt.% C, like steel A.

Regarding the fracture strength (FS) of the steels, the behavior is basically the same as the YS and the UTS. However, in the C steel, it is appreciated that the difference between the engineering UTS and the FS reaches only 16 MPa, which gives the first indication of a predominantly homogeneous strain throughout the test.

An important factor in steels with TRIP behavior is the homogeneous tensile elongation capacity. When reviewing the results of the uniform elongation (UE), both engineering and true, it is possible to verify that steels A, B, and C present similar percentages (11–12%). However, these same results when relating them to the UTS, steels B and C present a notable better behavior than steel A. Meanwhile, steel D presents an engineering and true UE, notably higher than steels A, B and C, close to 16%, with an engineering UTS like steel A, and in the case of true UTS, higher than steel A by 31 MPa, which could be attributable to the TRIP effect.

When reviewing the total elongation (TE), for steels A, B and C, as the wt.% C increases, the percentage of elongation decreases. Once again, a special case is represented by D steel, due to its remarkable TE of 22.3%, surpassing the TE of steel A by almost 4%.

Regarding area reduction (Exz), the behavior of the steels is similar and consistent with the ductility shown with TE and the wt.% C increase. A notable case is represented by steel C due to its almost zero Exz, showing signs of brittle behavior under traction. The case of D steel is also notable because, despite having the same chemical composition as A steel, it has 6.4% less Exz with higher elongations, a behavior that could be explained by the TRIP effect.

Finally, the strain hardening index (n), extracted from the Hollomon equation, which determines the slope of the true plastic strain and true stress curve, shows that for steels A, B and C, as it increases the wt.% C, n increases, being consistent with the increase in vol.% of RA and the possible TRIP effect. A notable and exceptional case is steel D, since despite being the one with the lowest wt.% C (same as A), it is the one that presents the highest n of the four, which confirms the most notorious TRIP effect, possibly due to the balance between its different heat treatment that provided a ductile PF matrix and the amount of RA involved in transformation.

### 3.3.2. Fracture Surfaces

Figure 10 show fracture surfaces by SEM at 1000× of magnification, after tensile test, to A, B, C and D steels in Figure 10a–d, respectively.

Steel A (Figure 10a) clearly shows a dimpled surface, typical of ductile fracture, characterized by equiaxed cup—like depressions, where the microvoids are initiated at second—phase particles, in this case possibly in M/RA particles (lath or block) over BF matrix [31].

Steel B (Figure 10b) shows a fracture surface that we could call quasi—cleavage, because the faces on the fracture surface are not true cleavage planes, since their size is much smaller than a prior—austenite grain. A mixture of ductile and brittle fracture is practically seen, with the presence of non-equiaxial dimples and signs of transgranular crack formation. The change in morphology with respect to steel A could be due to the increase in wt.% C, and consequently to the increase in vol.% of M/RA.

Steel C (Figure 10c) shows a cleavage fracture, typical of brittle behavior, occurring along crystallographic planes. It presents flat facets about the size of prior—austenite grain, exhibiting some "river markings", caused by the crack moving through the crystal along several parallel planes which form a series of plateaus and connecting ledges. In the center of the micrograph, a large transgranular crack can be seen, penetrating several previous austenitic grains. The morphology of this fracture surface can be attributed to a large amount of M/RA block microconstituent, which is predominantly brittle.

Steel D (Figure 10d) show a dimpled surface, typical of ductile fracture, characterized by equiaxed cup—like depressions, where the micro voids are initiated at second–phase particles, remarkably like A steel, but in this case, M/RA particles could have initiated the microvoids over a soft PF matrix [32,33].

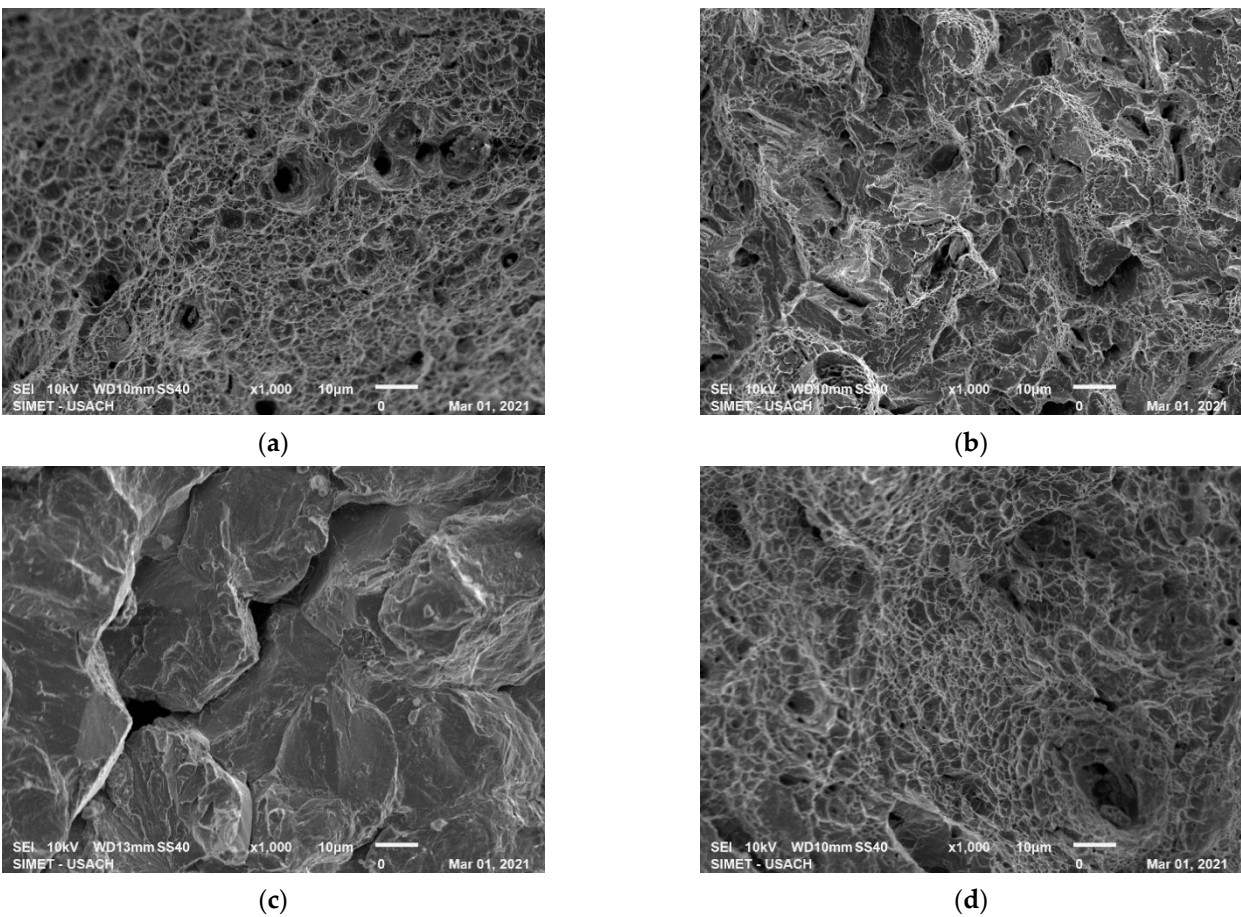

**Figure 10.** Fracture surface after tensile test. Results obtained for steels A, B, C, D by SEM at 1000× (**a**–**d**).

## 4. Discussion

The amount of carbon was decisive in the evolution of the microstructure achieved in steels A, B and C. Although these were subjected to the same heat treatment to achieve TBF steels, a clear difference was generated between them in terms of vol.% of RA, determined by XRD, and the morphology of its phases and microconstituents, observed by OM, SEM and AFM. As the wt.% C increased, the amount of BF decreased and the M/RA evolved from films to coarse blocks, perceiving an increase in its vol.% relative to BF. This is simply explained by the amount of carbon available in the high temperature austenite (910 °C), and that later, as the BF sheaves grow in the bainitic isothermal treatment (400 °C), a stable RA is generated at room temperature, rich in carbon, but free of carbides, in the zone adjacent to the BF sheaf, and residual M in the zones furthest from BF. Therefore, the more carbon available, the higher the vol.% M/RA and its morphology will tend to evolve from films to blocks. A different case was observed in steel D, since, although the chemical composition is the same as steel A, the heat treatment contemplated an IA at 760 °C, to achieve a steel with conventional TRIP or TPF behavior, where sought to obtain around 50% ductile FP. This determined that the 50% of austenite achieved in the IA was enriched with wt.% C between 0.5–0.6, higher than A, B and C, consequently obtaining a large amount of vol.% of RA (17.9%), despite having less austenite available for bainitic transformation, compared to other steels. Morphologically, D steel also showed a different microstructure, due to a granularization process of the BF and M/RA, atypical in the bainitic transformation with isothermal treatments, but not in the TPF steels obtained by continuous cooling. A possible explanation can be found in the fact that with a high wt.% C in austenite, Bs is lower than in the rest of the steels. This implies that at 400 °C, IT temperature, a noticeable "upper BF" is obtained, less fine than the one formed at lower temperatures. This phase was also aged after 1500 s of bainitic transformation, evolving from plates to granules. This also affects

the morphology of the M/RA, similar to what could occur in continuous cooling, such as explained in Section 3.2.1.

Regarding the techniques used for the identification of phases and microconstituents, to characterize TBF and TPF steels, it is necessary to highlight that they are all complementary. As shown in the analysis of results, none of the microscopy techniques used was effective enough to determine the specific presence of RA, however, they generated clarity in the identification of the M/RA microconstituent. The foregoing makes it clear that so far, the most efficient way to determine the presence of RA is XRD, with which its carbon enrichment can even be calculated approximately. In future studies, the use of EBSD (Electron Back Scattering Diffraction) for the determination of zones with crystallographic orientation related to RA (FCC), could be used to contrast the results obtained with XRD and obtain an image of the distribution of RA in the matrix of the steels. Notable is the fact that, with the same etch technique on the samples reasonable results could be obtained for OM, SEM and AFM, using only the concept of selective dissolution of phases and microconstituents by Nital. Although, it is not the best technique to perform an OM in polyphase steels, because it only distinguishes the ferritic grain boundaries, it generates the necessary conditions for the visualization of valleys and peaks with SEM and AFM. These techniques, due to their capacity to process images at high magnifications, were decisive for the observation of microstructures as fine as the BF and the M/RA in films and blocks. The AFM, given its potential to evaluate surfaces with atomic resolution, represents a great tool for future studies in the observation and determination of morphology for nanostructured polyphasic steels. Such is the case of ultrafast annealing steels currently under investigation.

Although the quantification of RA, its morphology and its stability against plastic strain is important to identify and characterize steels with TRIP behavior, it should not be forgotten that the goal of all heat treatment is to enhance the mechanical properties of the material. In the present study, the tensile test was chosen to relate the amount of carbon, the microstructure achieved, the strength, the strain, and the fracture surface, to evaluate the TRIP effect of the steels, by virtue of the different capacities that each one of them possesses to induce plasticity due to the martensitic transformation of RA. As established in the analysis of the results of Section 3.3.1, in steels A, B and C, as wt.% C increases, consequently the YS and UTS rises, and the TE decreases, due to the increase in volume and thickening of the M/RA microconstituent, which is hard and brittle. This is confirmed by the fracture surfaces, where an evolution from a ductile morphology (with multiple dimples) to a brittle morphology (characterized by cleavage, river marks and transgranular fracture). However, when comparing the UE, it is seen that steel C, despite being the most resistant and fragile, with a large amount of M/RA in blocks, has a similar and even higher UE than steels A and B, which have less carbon and less resistance. This clearly indicates a greater TRIP effect in C steel, due to its large amount of RA, and which is also evidenced by its higher strain hardening index "n". On the other hand, due to the IA, the D steel becomes TPF, that is, it has around 50 vol.% of PF, which implies that the YS is lower than the TBF steels, due to the lower density of dislocations in the matrix phase (PF versus BF) and the differences in wt.% C. Subsequently, when relating its UTS with the UE (engineering), a noticeable TRIP effect is appreciated, due to the marked increase in both, up to 897 MPa of resistance and 16% of elongation, which is also confirmed in the true curve with the calculation of the index "n", a parameter that is far superior to TBF steels. The initial and final behavior of steel D is ductile, managing to harden as it progresses in homogeneous plastic deformation, reaching a UTS like that of steel A, but with a widely higher UE, which is confirmed too by analyzing the surface of fracture, which presents an eminently ductile morphology.

## 5. Conclusions

- The effect of carbon was decisive in the evolution of the microstructure achieved in steels A, B and C. These were subjected to the same heat treatment to achieve TBF

steels, developing between them a difference in terms of vol.% of RA, and morphology of its phases and microconstituents, which were observed by OM, SEM and AFM. As the wt.% C increased, the amount of BF decreased and the M/RA evolved from sheets to increasingly thicker blocks, perceiving an increase in its vol.% relative to BF.

- Steel D showed a granularization process of the BF and M/RA, atypical in the bainitic transformation with isothermal treatments. A possible explanation could be that with a high wt.% C in austenite, the Bs is lower than in the rest of the steels, implying that at 400 °C, IT temperature, a higher BF is obtained, less fine than the one formed at lower temperatures, which was subsequently aged after the 1500 s of bainitic transformation, evolving from plates to granules, also affecting the morphology of the M/RA, like what could occur in continuous cooling.

- Given the potential to evaluate surfaces with atomic resolution, AFM is a tool for future studies in the observation and determination of morphology for nanostructured polyphase steels, achieved with modern techniques of heat treatments for fine grain size, such as ultrafast annealing.

- In steels A, B and C, as the percentage of carbon increases, the YS and UTS rise, and the TE decreases, due to the increase in volume and thickening of the M/RA microconstituent, which is hard and brittle, confirmed with the fracture surfaces, showing an evolution from a ductile to a brittle morphology.

- Steel C, despite being the strongest and most brittle, with a large amount of M/RA in blocks, has a similar and even higher UE than steels A and B, which have less carbon and less strength. The above indicates a greater TRIP effect, due to the large amount of RA, which is evidenced by its higher "n" index.

- Steel D is a TPF steel achieved with an IA. The initial and final behavior is ductile, hardening as it progresses in homogeneous plastic strain, reaching a UTS like of steel A, but with a widely higher UE, which is confirmed too by analyzing the surface of fracture, which presents a ductile morphology.

- From the point of view of tensile toughness, the steel with the best characteristics to be used in the high-thickness structural field is D. Despite having the lowest YS, it has the best YS/UTS ratio, the highest UE, the largest "n", and its fracture mode is mainly ductile. The higher toughness is attributed to the soft and ductile PF matrix with fine aggregates of M/RA dispersed throughout the matrix in the form of granules, which concentrate less stress than acicular morphology.

- The most important aspect of this work is that the best mechanical behavior of the high thicknesses TRIP steels studied, is related to a fine morphology of the microstructure and an adequate amount of retained austenite.

**Author Contributions:** Conceptualization, E.T. and A.M.; data curation, E.T.; formal analysis, E.T.; funding acquisition, A.M.; investigation, E.T.; methodology, E.T.; project administration, A.M.; resources, A.A., C.S. and A.M.; software, E.T. and C.S.; supervision, A.M., A.A., C.S.; validation, E.T. and A.A.; visualization, E.T.; writing—original draft, E.T.; writing—review and editing, E.T., A.A. and A.M. All authors have read and agreed to the published version of the manuscript.

**Funding:** This research received no external funding.

**Institutional Review Board Statement:** Not applicable.

**Informed Consent Statement:** Not applicable.

**Data Availability Statement:** Data is contained within the article. The data presented in this study are available in Ph. D. Thesis of Enzo Tesser entitled "Influence of the microstructure on the fracture toughness and other mechanical properties of a FeCMnSiCr ferritic-bainitic steel with TRIP behavior", USACH 2021, Santiago, Chile.

**Acknowledgments:** Enzo Tesser gratefully acknowledges the support provided by "SIMET USACH" of Universidad de Santiago de Chile. Alberto Monsalve gratefully acknowledges the support of DICYT USACH, Grant 052014MG. Carlos Silva gratefully acknowledges the support of FONDECYT INICIACION N°11190782 and FONDEQUIP-EQM 150106.

**Conflicts of Interest:** The authors declare no conflict of interest.

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
