# Peer review of "Effect of Carbon Content and Intercritical Annealing on Microstructure and Mechanical Tensile Properties in FeCMnSiCr TRIP-Assisted Steels"

_metals, doi:10.3390/met11101546_

Round 1

Reviewer 1 Report

Car manufacturers are trying to reduce the weight of new models due to increasingly stringent regulations on carbon dioxide emissions. The most effective way reduce to weigh the body of cars is, above all, used the hight strength steel sheets, and used of sheets with a smaller thickness  of its individual parts . From this point of view, the topic is very topical.

In the introduction, the authors paid considerable attention to the development of advanced high-strength TRIP steels. However, they did not pay due attention to the effect of alloying elements on the TRIP effect.

Comment: Add knowledge about the influence of microalloying (influence of alloying elements) on TRIP effect!

When forming steels with the TRIP effect, their strength increases, which provides designers with new possibilities.

Parts of automobiles, ships, which are to absorb impact energy in the event of an impact, are produced from these steels by forming.

It is also necessary to take into account the effect of the TRIP effect on the formability.

Formability, among other factors, also depends on the scheme stress and strain pattern, i. sheet metal parts can be made by deep drawing, stretching, etc.

The effect of the TRIP effect may be different for individual sheet metal stamping operations.

Comment: Add knowledge about the TRIP effect on formability and about evaluating the formability of steel sheets based on their mechanical properties!

This section describes in detail the materials used, heat treatment procedures, structural and microstructural analysis….

Add: Why was a 7 mm thick material used? TRIP steel sheets for car parts are most often used up to a thickness of approx. 3 mm. What influence does the thickness of the used sheet have on the choice of the heat treatment procedure and mechanical properties resp. their utility properties?

Add: Used equipment and its possibilities for obtaining different states of steels (S1, S2, S3).

Steels with a TRIP effect are characterized by a very good energy absorption capacity in the event of an impact. The ability to absorb energie during deformation can be expressed as a function of the true stress na true strain. Therefore, it is necessary to supplement Hollomon's relationship to determine the true stress.

Figure 9 does not correctly show the deformations in% (for example 12% but 0.12 or the true - logarithmic deformation). In order to assess accuracy, it is necessary to supplement the relationships used to determine engineering stress and true stress.

Add resp. Edit: In table 3 it is necessary to clearly state the symbols for engineering stress and true stress from the tensile test. If the exponent of strain hardening was evaluated using the Hollomon relation, then in the table state the values ​​of the constant K (Strength Coefficient) found for materials S1, S2, S3.

In conclusion, it would be appropriate to state which of the materials is most suitable for application on a car body parts

Author Response

Comments of Reviewer 1

Car manufacturers are trying to reduce the weight of new models due to increasingly stringent regulations on carbon dioxide emissions. The most effective way reduce to weigh the body of cars is, above all, used the hight strength steel sheets, and used of sheets with a smaller thickness of its individual parts. From this point of view, the topic is very topical.

In the introduction, the authors paid considerable attention to the development of advanced high-strength TRIP steels. However, they did not pay due attention to the effect of alloying elements on the TRIP effect.

Comment: Add knowledge about the influence of microalloying (influence of alloying elements) on TRIP effect!

ANSWER or COMMENT: Knowledge about influence of alloying elements on TRIP effect was included in introduction section.

The following text was added:

If the TBF steels are applied to relatively large forging parts, high hardenability may be required to obtain the mixed microstructure of bainitic ferrite and metastable retained austenite. In general, hardenability of the steel is improved by the addition of alloying elements such as Cr, Mo, Ni, Mn, B, etc. However, there is no research investigating the effects of hardenability on microstructure and mechanical properties in the hot-forged medium-carbon TBF steels.

When forming steels with the TRIP effect, their strength increases, which provides designers with new possibilities.

Parts of automobiles, ships, which are to absorb impact energy in the event of an impact, are produced from these steels by forming.

It is also necessary to take into account the effect of the TRIP effect on the formability.

Formability, among other factors, also depends on the scheme stress and strain pattern, i. sheet metal parts can be made by deep drawing, stretching, etc.

The effect of the TRIP effect may be different for individual sheet metal stamping operations.

Parts of automobiles, ships, which are to absorb impact energy in the event of an impact, are produced from these steels by forming.

Comment: Add knowledge about the TRIP effect on formability and about evaluating the formability of steel sheets based on their mechanical properties!

ANSWER or COMMENT: unlike in the automotive industry, given the thickness of the sheets, in shipbuilding the most of structural parts and pieces are not manufactured by formability (stamping, deep drawing) and not in this type of steels, just some hull parts are manufactured by forging (doubly curved). Although formability is not relevant in heavy industry, knowledge about the TRIP effect on formability and about evaluating the formability of steel sheets based on their mechanical properties was included in introduction section.

The following text was added:

TRIP effect increases the homogeneous strain, so it is expected acceptable formability. Nevertheless, most important in formability of metals and alloys are the normal anisotropy index r (always known as Lankford coefficient) and the planar anisotropy index (Dr). However, the focus of this work is heavy industry, where formability is not the mean desired mechanical property.

Add: Why was a 7 mm thick material used? TRIP steel sheets for car parts are most often used up to a thickness of approx. 3 mm. What influence does the thickness of the used sheet have on the choice of the heat treatment procedure and mechanical properties resp. their utility properties?

ANSWER or COMMENT: The thickness used for this investigation was not 7 mm, but 15 mm. The reason is described in the introduction section, since the objective and the novelty of this research is to study the TRIP behavior in plates of greater thickness than those used in the automotive industry, with the purpose of giving a use in other steel consuming industries where high strength and toughness of steels are desirable. Therefore, as a first step in that sense, the evolution of its microstructure and tensile mechanical properties were evaluated. This explanation was reinforced in the introduction section.

Steels with a TRIP effect are characterized by a very good energy absorption capacity in the event of an impact. The ability to absorb energie during deformation can be expressed as a function of the true stress na true strain. Therefore, it is necessary to supplement Hollomon's relationship to determine the true stress.

Figure 9 does not correctly show the deformations in% (for example 12% but 0.12 or the true - logarithmic deformation). In order to assess accuracy, it is necessary to supplement the relationships used to determine engineering stress and true stress.

Add resp. Edit: In table 3 it is necessary to clearly state the symbols for engineering stress and true stress from the tensile test. If the exponent of strain hardening was evaluated using the Hollomon relation, then in the table state the values of the constant K (Strength Coefficient) found for materials S1, S2, S3.

ANSWER or COMMENT: In figure 9, the engineering graphs (figure 9.a) and the true ones (figure 9.b) are shown. The true graphs were obtained only up to the UTS, in order to work with the homogeneous strain and to be able to determine more precisely the strain hardening index, according to the Hollomon setting. Table 5 shows the results, both engineering and true, of the tensile test. In the true section of the table, the index "n" is included for each curve and the constant K is missing, which was included for steels A, B, C and D.

The following text was added:

The relationships used to compute true stress and true strain were:

where  is the engineering strain;  is the true strain;  is the engineering stress and  is the true stress.

In conclusion, it would be appropriate to state which of the materials is most suitable for application on a car body parts

ANSWER or COMMENT: Although the main objective of this research is not focused on applications for automotive parts, the material that presents the best characteristics to be used in the automotive field is D, due to its strength, amount of homogeneous elongation and strain hardening index.

Reviewer 2 Report

In this work three Bainitic Ferrite TRIP steels and one Polygonal Ferrite TRIP steel with three different carbon content (0.2; 0.3 and 0.4 wt.% C) were studied to study the evolution of their microstructure and tensile mechanical properties. I have summarized some points regarding this work below:

  • The abstract is written well. Maybe mention what TRIP stands for in Abstract.
  • The intro on TRIP, TPF and TBF is well written and concise.
  • The methodology description is clear. However, it would be beneficial if the authors add a word on the error associated with the XRD-based methods they used to measure the austenite vol. fraction and carbon content measurements in austenite. Especially, since the entire discussion revolves around the effect of carbon content I RA, it would be beneficial to use a complimentary method that is more accurate to measure the carbon content.
  • In the results section, characterization of the constituent phases using optical and SEM , especially M/RA, seems weak, and requiring back-up using methods such as TEM analysis. For example what is labelled as M/RA block in Fig 6(a), seems to be BF with very fine lath structure. The designation of PF and BF in general are not completely backed-up. Though the AFM results are interesting and noteworthy, unfortunately the AFM results do not remove the ambiguity either, as the phase identification has been done purely based on the topography of Nital etched samples.
  • The fractography results are sound and well presented.
  • Some of the sentences in the discussion are too long and need to be broken into smaller sentences for better clarity. e.g. lines 450-455; 458-460; 488-491
  • In line 459, and also 497, the word "notoriously" needs to be replaces by a more appropriate word.
  • In line 492, change "strength" to "strong".
  • All in all this study is interesting but weak in terms of methodology used for phase identification and carbon content measurement which are the basis of the entire discussion. On the other hand, the conclusions are not clear and do not offer a solid take-away that is applicable to the science of TRIP steels or the industry. I suggest a paragraph be added to the conclusions on the significance of this work to the science of designing TRIP steels. 

Author Response

Comments of Reviewer 2

The abstract is written well. Maybe mention what TRIP stands for in Abstract.

ANSWER or COMMENT: it was mentioned in abstract.

The methodology description is clear. However, it would be beneficial if the authors add a word on the error associated with the XRD-based methods they used to measure the austenite vol. fraction and carbon content measurements in austenite. Especially, since the entire discussion revolves around the effect of carbon content I RA, it would be beneficial to use a complimentary method that is more accurate to measure the carbon content.

ANSWER or COMMENT: the error associated with the XRD based methods we used to measure the austenite volume fraction and carbon content measurements in austenite was mentioned in results section. This error is around 3-4%. The discussion does not revolve around the amount of carbon present in the retained austenite, but rather on the evolution that it could undergo the morphology of the microstructure and the repercussions on the tensile mechanical properties. In future research, a more precise method could be used for the determination of carbon in retained austenite, such as atom probe, however in this instance it was only possible to use x-ray diffraction for the determination of its lattice parameter, which is traditionally used to determine the amount of carbon.

Related to C content, the method takes into account the precision associated to the determination of angles. In accordance to reference 19, the precision is +/-0.005º.

In the results section, characterization of the constituent phases using optical and SEM , especially M/RA, seems weak, and requiring back-up using methods such as TEM analysis. For example what is labelled as M/RA block in Fig 6(a), seems to be BF with very fine lath structure. The designation of PF and BF in general are not completely backed-up. Though the AFM results are interesting and noteworthy, unfortunately the AFM results do not remove the ambiguity either, as the phase identification has been done purely based on the topography of Nital etched samples.

ANSWER or COMMENT: a technique such as TEM would certainly be enriching for the characterization of the microstructure and support the SEM. However, in this research the technique to support the SEM is AFM, which allows obtaining better resolutions than TEM and generating a 3D image, being able to even evaluate the different heights of the microconstituents involved, solving between PF, BF and M / RA, representing a good way to characterize multiphase steels with relatively simple methods, being one of the objectives of this research. For example, M/RA block in Fig 6(a) can be identified by relative height in comparison with BF. However, based on what is suggested by the reviewer, TEM will be applied in the coming months in order to define the different phases present.

Some of the sentences in the discussion are too long and need to be broken into smaller sentences for better clarity. e.g. lines 450-455; 458-460; 488-491

In line 459, and also 497, the word "notoriously" needs to be replaces by a more appropriate word.

In line 492, change "strength" to "strong".

ANSWER or COMMENT: it was changed in the manuscript.

All in all this study is interesting but weak in terms of methodology used for phase identification and carbon content measurement which are the basis of the entire discussion. On the other hand, the conclusions are not clear and do not offer a solid take-away that is applicable to the science of TRIP steels or the industry. I suggest a paragraph be added to the conclusions on the significance of this work to the science of designing TRIP steels.

ANSWER or COMMENT: The discussion does not revolve around the amount of carbon present in the retained austenite, but rather on the evolution that it could undergo in the morphology of the microstructure and the repercussions on the tensile mechanical properties. In fact, the heat treatments that produced the highest carbon enrichment in the retained austenite of all the steels were chosen. The most important aspect of this study is the amount of retained austenite and the morphology of the different phases and microconstituents that allow to conclude the subsequent mechanical behavior of the steels, which have thicknesses of 15 mm, applicable to steel consuming industries other than automotive. This explanation was developed in the conclusions like significance of this work to the science of designing TRIP steels.

In the introduction, the authors paid considerable attention to the development of advanced high-strength TRIP steels. However, they did not pay due attention to the effect of alloying elements on the TRIP effect.

ANSWER or COMMENT: it was added in the introduction section.

Reviewer 3 Report

  1. Knowledge of effect of carbon content and intercritical annealing in TRIP steels is known. Please describe precisely the aim and novelty of the paper in relation to the current state of the art.
  2. What was the criterion for choosing steel chemical compositions?
  3. The dilatometric investigations should be performed to determine the A1, A3 and all other critical temperatures.
  4. A precise description of the thermal treatment cycles must be added.In intercritical annealing. It is very important to provide both heating and cooling rates.
  5. Steel markings are too complicated for readers (S1, S2, S3, A, B, C, D).
  6. What the authors understand the concept of optimization of mechanical properties in the discussion of results?
  7. Please change the conclusions paying a special attention on the new findings of the paper in reference to the state of the art. Moreover, conclusions have to be modified. In the current state these do not inform about the new knowledge as an effect of the investigations. A few of them can not be considered as conclusions, e.g.

-“None of the microscopy techniques used was effective enough to determine the specific presence of RA…” RA can be analyzed using EBSD method which was not applied in the paper.”

- “With the same etch technique on the samples, excellent results could be obtained for OM, SEM and AFM, using only the concept of selective dissolution of phases and microconstituents by Nital”. What do the excellent results mean?

Author Response

Comments of Reviewer 3

Knowledge of effect of carbon content and intercritical annealing in TRIP steels is known. Please describe precisely the aim and novelty of the paper in relation to the current state of the art.

ANSWER or COMMENT: the novelty of this research is to study the TRIP behavior in plates of greater thickness than those used in the automotive industry, with the purpose of giving a use in other steel consuming industries, where high toughness steels are required. Therefore, as a first step in that sense, the evolution of its microstructure and tensile mechanical properties were evaluated. This explanation will be reinforced in the introduction section.

What was the criterion for choosing steel chemical compositions?

ANSWER or COMMENT: First, starting from a typical chemical composition of TRIP steels, the carbon content was increased in order to evaluate its effect on the morphology of the different phases and micro-constituents achieved by different heat treatments. Second, alloying elements, mainly Chromium, was added in order to achieve a better hardenability of the steels, since one of the objectives of this research was to work with 15 mm thick plates, oriented to use in industries other than the automotive one. These ideas were included in the text (in Introduction).

The dilatometric investigations should be performed to determine the A1, A3 and all other critical temperatures.

 ANSWER or COMMENT: clearly the dilatometer technique is the most accurate for obtaining critical temperatures. Since this instrument was not available for this work, it was decided to use the DTA, DSC, bibliographic analysis, and image analysis techniques in a complementary manner. We believe that we reached satisfactory results.

A precise description of the thermal treatment cycles must be added.In intercritical annealing. It is very important to provide both heating and cooling rates.

ANSWER or COMMENT: it was added in experimental procedure section.

Steel markings are too complicated for readers (S1, S2, S3, A, B, C, D).

ANSWER or COMMENT: We agreed, but it was the best way to make the difference between steels developed only by chemical composition and those achieved with subsequent heat treatment.

What the authors understand the concept of optimization of mechanical properties in the discussion of results?

ANSWER or COMMENT: the word was changed by one better.

Please change the conclusions paying a special attention on the new findings of the paper in reference to the state of the art. Moreover, conclusions have to be modified. In the current state these do not inform about the new knowledge as an effect of the investigations. A few of them cannot be considered as conclusions, e.g.

o          -“None of the microscopy techniques used was effective enough to determine the specific presence of RA…” RA can be analyzed using EBSD method which was not applied in the paper.”

o          - “With the same etch technique on the samples, excellent results could be obtained for OM, SEM and AFM, using only the concept of selective dissolution of phases and microconstituents by Nital”. What do the excellent results mean?

ANSWER or COMMENT: The conclusions were modifed on paying special attention to the development of steels with a TRIP effect of different chemical compositions with a thickness of 15 mm, different from those traditionally used in the automotive industry and that could be used in heavy industries. The conclusions given as an example will be eliminated.

Round 2

Reviewer 2 Report

I'd like to thank the authors for considering the comments made on the manuscript.

The use of AFM as a methodology for phase identification is still problematic. Answering my second comments the Authors wrote: "which allows obtaining better resolutions than TEM and generating a 3D image, being able to even evaluate the different heights of the microconstituents involved, solving between PF, BF and M / RA, representing a good way to characterize multiphase steels with relatively simple methods, being one of the objectives of this research."

The authors claim that AFM could resolve nonmetric features better than TEM. This is a big claim and needs a separate research dedicated to it. While the AFM relies on the topography of the etched surface, and SEM relies on the morphology/topography. In SEM, variation in the height of the features does not create ambiguity in interpreting the features, while in AFM it does. It might be true that different phases react differently to the etchant, but relating the height of the features to certain phases it is not completely scientific, and needs further proof/discussion. On the other hand if exploring the AFM was in fact one of the objectives of this work, it would have been mentioned in the title of the work or at least in the abstract. The AFM method is not and should not be part of the objectives. My suggestion is it should be further toned down or even removed as in reality removing the AFM section does not change anything in the paper and does not affect any of the main conclusions. I would suggest if the authors would like to show the capabilities of AFM for phase identification in steels, or compare its capabilities to TEM, they write a separate characterization paper on that subject.

There are still errors and typos in the text that need attention. Some of them are highlighted in the attached manuscript file.

Author Response

Comments of Reviewer 2 (round 2)

I'd like to thank the authors for considering the comments made on the manuscript.

The use of AFM as a methodology for phase identification is still problematic. Answering my second comments the Authors wrote: "which allows obtaining better resolutions than TEM and generating a 3D image, being able to even evaluate the different heights of the microconstituents involved, solving between PF, BF and M / RA, representing a good way to characterize multiphase steels with relatively simple methods, being one of the objectives of this research."

The authors claim that AFM could resolve nonmetric features better than TEM. This is a big claim and needs a separate research dedicated to it. While the AFM relies on the topography of the etched surface, and SEM relies on the morphology/topography. In SEM, variation in the height of the features does not create ambiguity in interpreting the features, while in AFM it does. It might be true that different phases react differently to the etchant, but relating the height of the features to certain phases it is not completely scientific, and needs further proof/discussion. On the other hand if exploring the AFM was in fact one of the objectives of this work, it would have been mentioned in the title of the work or at least in the abstract. The AFM method is not and should not be part of the objectives. My suggestion is it should be further toned down or even removed as in reality removing the AFM section does not change anything in the paper and does not affect any of the main conclusions. I would suggest if the authors would like to show the capabilities of AFM for phase identification in steels, or compare its capabilities to TEM, they write a separate characterization paper on that subject.

ANSWER or COMMENT: When we talked about objectives, we were referring to the ability to characterize a multiphase material, not to the AFM technique only. The synergy between optical microscopy, SEM and AFM allowed to carry out a morphological and topographic evaluation of the microstructure and to relate it to the mechanical properties. For example, when trying to distinguish between PF and BF in D steel, SEM does not succeed, while AFM does due to its topographic resolution. We agree that the comparison between TEM and AFM could be analyzed in depth in a different job, what's more, we believe that it is an excellent idea for further work on multiphase steels.

There are still errors and typos in the text that need attention. Some of them are highlighted in the attached manuscript file.

ANSWER or COMMENT: we agree, the errors were corrected in the manuscript.
